# Comparison of double chevron-cut and biplanar distal femoral osteotomy techniques: A biomechanical study

Kuan-Jung Chen[1,2], Wei Hsiung[2,3], Chien-Yuan Wang[1,4], Oscar Kuang-Sheng Lee[4,5,6,7], Kuo-Kuang Huang[8], Ye Huang[9], Jesse Chieh-Szu Yang[2,5,10]*

1 Department of Orthopaedics, China Medical University Hsinchu Hospital, Hsinchu, Taiwan, 2 Department of Orthopaedics, School of Medicine, National Yang Ming Chiao Tung University, Taipei, Taiwan, 3 Department of Orthopaedics, Shin Kong Wu Ho-Su Memorial Hospital, Taipei, Taiwan, 4 Department of Orthopaedics, College of Medicine, China Medical University, Taichung, Taiwan, 5 Institute of Clinical Medicine, National Yang Ming Chiao Tung University, Taipei, Taiwan, 6 Center for Translational Genomics and Regenerative Medicine, China Medical University Hospital, Taichung, Taiwan, 7 Stem Cell Research Center, National Yang Ming Chiao Tung University, Taipei, Taiwan, 8 Department of Information Science, National Penghu University of Science and Technology, Penghu, Taiwan, 9 Department of Orthopedics, Knee Preservation Clinical and Research Center, Beijing Jishuitan Hospital, Beijing, China, 10 Department of Orthopaedics and Traumatology, Taipei Veterans General Hospital, Taipei, Taiwan

* jeffyang80@hotmail.com

**Data Availability Statement:** All relevant data are within the manuscript and its Supporting Information files.

## Abstract

### Objective

This study aimed to compare the stability and mechanical properties of the double chevron-cut (DCC) and biplanar (BP) distal femoral osteotomy (DFO) techniques, along with analyzing their respective contact surface areas.

### Methods

Biomechanical testing was performed using sawbone and 3D modeling techniques to assess axial and torsional stability, torsional stiffness, and maximum torque of both osteotomy configurations. Additionally, 3D models of the sawbone femur were created to calculate and compare the contact surface area of the DCC, BP, and conventional single-plane DFO techniques.

### Results

Axial stiffness and maximum strength did not significantly differ between the two osteotomy techniques. However, in terms of torsional properties, the DCC technique exhibited superior torsional stiffness compared to the BP group (27 ± 7.7 Nm/° vs. 4.5 ± 1.5 Nm/°, p = 0.008). Although the difference in maximum torque did not reach statistical significance (63 ± 10.6 vs. 56 ± 12.1, p = 0.87), it is noteworthy that the DCC group sawbone model exhibited fracture in the shaft region instead of at the osteotomy site. Therefore, the actual maximum torque of the DCC construct may not be accurately reflected by the numerical values obtained in this study. The contact surface area analysis revealed that the BP configuration had the largest contact surface area, 111% larger than that of the single-plane configuration. but

**Funding:** The authors received no specific funding for this work.

**Competing interests:** The authors have declared that no competing interests exist.

60% of it relied on the less reliable axial cut. Conversely, the DCC osteotomy offered a 31% larger contact surface area than the single-plane configuration, with both surfaces being weight-bearing.

## Conclusion

The DCC osteotomy exhibited superior mechanical stability, showing improved rotational stiffness and maximum torque when compared to the BP osteotomy. Although the BP osteotomy resulted in a larger contact surface area than the DCC osteotomy, both were larger than the conventional single-plane configuration. In clinical practice, both the DCC and BP techniques should be evaluated based on patient-specific characteristics and surgical goals.

## Introduction

Distal femoral osteotomy (DFO) is an established surgical technique for the treatment of various knee pathologies, such as genu valgum deformity, osteoarthritis in the lateral compartment, and distal femoral rotational deformity with recurrent patellar subluxation [1–3]. However, achieving primary stability to allow immediate weight bearing has remined challenging.

The biplanar (BP) DFO technique was introduced by Freiling et al to improve the stability and contact surface of the construct [4]. This technique aimed to increase primary stability of the osteotomy and prevent rotational malalignments. However, the clinical and biomechanical outcomes of BP osteotomy are controversial. Studies found that the torsional stability of BP DFO was less than that of single plane DFO [5], and the anterior flange of the BP osteotomy was prone to fracture [6,7]. Furthermore, the studies by Nakayama et al. and Kambara et al. showed that BP DFO generates internal rotation and flexion due to the mismatch of the two transverse cuts [8,9].

Our previous study had proposed a novel double chevron-cut (DCC) osteotomy to address the limitations of the BP osteotomy [10]. The technique involves making two V-shaped cuts in the distal femur, creating two chevron-shaped bone fragments that can be aligned and stabilized with a locking plate. Other than possible mechanical advantages, the DCC technique bears the potential to place the osteotomy site more distally into the area of cancellous bone in the femur, a region with better healing potential. However, a limitation of our previous study was the lack of biomechanical simulation to further evaluate the performance of the DCC technique.

To the best of our knowledge, there is no comparative study on biomechanics between DCC technique and BP DFO technique. Here we investigate the stability and stiffness of DCC technique and BP DFO technique under axial or rotational load. The impact of cutting site and contact surface area of three different osteotomy techniques–including single plane, DCC, and BP–are discussed. We hypothesize that the DCC technique will demonstrate superior mechanical stability and stiffness compared to the BP DFO technique under both axial and rotational loads. Additionally, we expect to observe differences in the cutting site and contact surface area among the three osteotomy techniques (single plane, DCC, and BP).

## Materials and methods

For our biomechanical tests, we utilized medium-sized fourth-generation synthetic analogue femur models (model #3403; Sawbones, Pacific Research Laboratories, Inc., Vashon, WA).

The model's specifications include a total length of 455 mm, a neck-shaft angle of 135˚, a shaft diameter of 27 mm, and a solid foam cancellous core with a density of 17 pounds per cubic foot (17 PCF) and a 13 mm canal diameter [11]. The selection of this model was based on its mechanical similarity to human bone properties [12–14]. A total of 12 femur sawbone models were used to compare the biomechanics of 2 osteotomy techniques under 2 different loading protocols, respectively. This study exclusively focuses on biomechanical research utilizing a synthetic bone model for testing, along with 3D model simulation. Consequently, ethical approval is not sought due to the absence of human participants.

## Osteotomy techniques

Two different configurations of medial distal femur close-wedge osteotomy were tested on the synthetic bones: (1) DCC DFO (10) and (2) BP DFO. The angle between the two planes of each osteotomy of were set to 110˚. Commercialized 3D-printed polyamide cutting guide jigs (A Plus Biotechnology Co., Ltd., New Taipei City, Taiwan) were made for both osteotomy techniques.

Each cutting jig was attached to a synthetic bone by four K-wires before wedge resection. After removal of the wedge, closing of the osteotomy was performed manually. One 8-hole distal medial femur osteotomy locking plate (A Plus Biotechnology Co., Ltd., New Taipei City, Taiwan) was used as fixation implant. These anatomically contoured locking plates were composed of titanium alloy (Ti-6Al-4V ELI), measuring 17 mm in width and 121 mm in length, with locking screws of 5.0 mm diameter. For proximal fixation, three bicortical and one unicortical screws were utilized, while four unicortical screws were used for distal fragment fixation. All screws were tightened with 4-Nm torque limiter in adherence with the standard surgical protocol. (Fig 1) All femur osteotomies and plate fixation were performed by a senior orthopaedic surgeon (Jesse C.S. Yang).

## Experimental set-up

A total of 12 in vitro experiments were conducted, with 3 replicates for each configuration tested for axial compression and torsional loading. The femur head, trochanters, and the distal femur condyles were embedded in polyurethane-based blocks in the fixturing vises, and the femur specimens were mounted on the testing machine (Instron E10000, Canton, MA, USA) via the vise in a valgus position, which closely replicated the normal weight-bearing orientation of the lower limb. To achieve this alignment, great care was taken to ensure that the line of action for the axial compression load passed precisely through the center of the femoral head and the intercondylar notch, simulating the mechanical axis of the femur. (Fig 2) The Instron E10000 system boasts a linear dynamic capacity of 10 kN, torsional dynamic capacity of 100 Nm, and offers a weighing accuracy of ±0.5% for both load and torque measurements. Data on load and deflection, as well as torque and rotation angle, were acquired and recorded using the Instron E10000 system.

## Loading protocol

The mechanical testing protocol followed the previous investigations by Brinkman et al [5,15], which were designed to simulate physiological conditions. Axial loading protocols were tested on three specimens of each osteotomy configuration. The femurs were subjected to cyclical axial loads with 150 and 800 N simulating partial and full weight-bearing in an adult of normal body weight of 80 kg. A compression axial 10 N preload was applied to remove slack, after which the femurs were tested to 100 cycles for each axial load at a frequency of 0.5 Hz. Subsequently, each femur was loaded to failure at a rate of 0.1 mm/sec. The load was increased incrementally until failure. Failure was defined as a drop in load due to failure of the construct, the

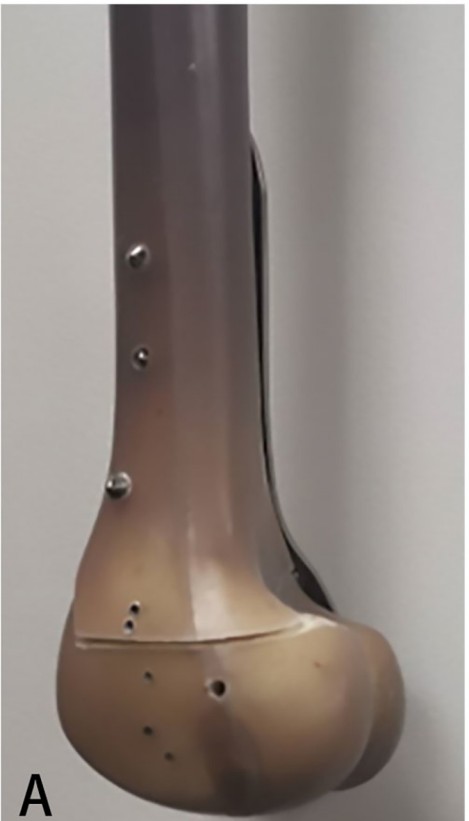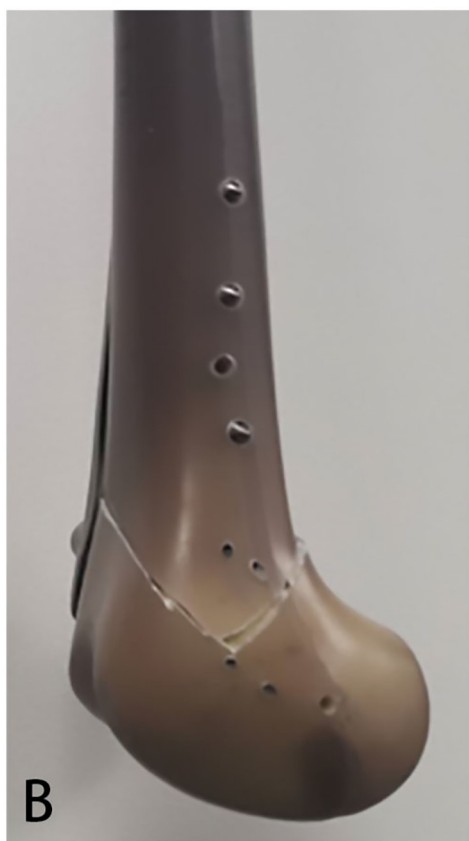

**Fig 1. Sawbone models.** Sawbone models of the two different configurations of medial distal femur close-wedge osteotomy (A) Biplanar Distal Femur Osteotomy (BP DFO) and (B) Double Chevron-Cut DFO (DCC DFO).

replicate bone, or the implant. The maximum strength was determined by progressive increased loading to failure. Axial construct stiffness was defined as the slope of the linear portion of the force-construct deformation curve.

Cyclical torsional loading was performed in three femurs per osteotomy configuration with a preload of 0.5 Nm. Cyclically torque of 5 Nm was applied at a rate of 0.25 HZ in internal rotation mode until 100 cycles were reached. During 100 cycles, an increasing axial preload was used to simulate no (0 N), partial (150 N), and full (800 N) weight-bearing. After completion of all three torsional loading tests, each femur was torqued to failure in a displacement control mode at 0.25˚/sec. The failure criterion and the determination of the mechanical properties were the same as used for the axial loading test. The maximum torque (Nm) was determined by applying torque until failure. The torsional stiffness was determined by plotting the rotation around the Z-axis over time against the moment applied by the material testing system. An overview of the test protocols has been summarized in Table 1.

## Contact surface area in different DFO constructs

In addition to the biomechanical testing, we conducted an additional analysis to compare the contact surface area of the three different osteotomy techniques: DCC, BP, and single-plane. 3D surgical planning models of the replicate femur were created, which were used to develop 3D-printed polyamide cutting guides. With these 3D models, the three different osteotomies were simulated, and the contact surface area for each technique was calculated and compared.

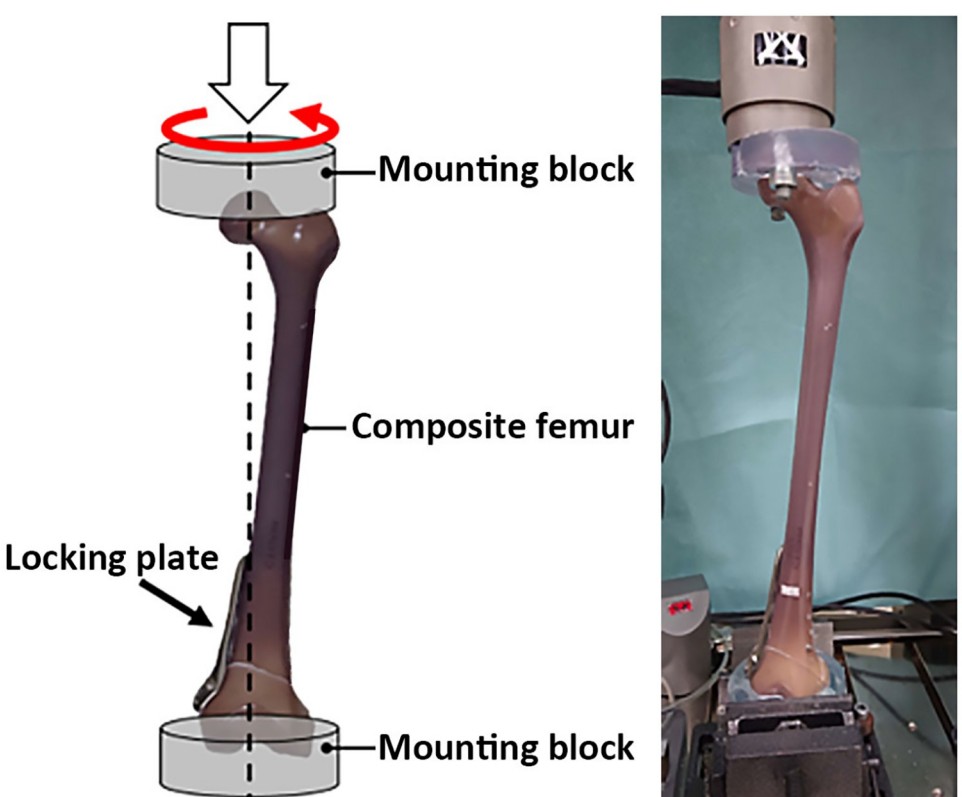

**Fig 2. The test set-up used in the study.** On the left, a schematic diagram depicts the configuration, while on the right, a corresponding photograph is displayed. The sawbone femur is subjected to loading using an Instron E10000 testing machine (Instron, Canton, MA, USA).

## Statistical analysis

Mechanical parameters are presented as the mean ± SD. The means and standard deviations of the construct stiffness and the maximum strength were calculated for the two configurations. Data normality was verified using the Shapiro-Wilk test, after which an independent sample t-test was employed for statistical comparisons. Statistically significant differences were defined by p-values of $< 0.05$. Statistical analysis was carried out using SPSS 20.0 statistical software (SPSS Inc., Chicago, IL).

**Table 1. Overview of the test protocols.**

| **Axial Loading** | | | | | |
|---|---|---|---|---|---|
| Test Sequence | Number of cycles | Axial preload (N) | Axial Load (N) | Torsional preload (Nm) | Torsional Load (Nm) |
| 1 | 100 | 10 | 150 | 0 | 0 |
| 2 | 100 | 10 | 800 | 0 | 0 |
| 3 | 1 | 0 | To failure | 0 | 0 |
| **Torsional Loading** | | | | | |
| Test Sequence | Number of cycles | Axial preload (N) | Axial Load (N) | Torsional preload (Nm) | Torsional Load (Nm) |
| 1 | 100 | 0 | 0 | 0.5 | 5 |
| 2 | 100 | 150 | 0 | 0.5 | 5 |
| 3 | 100 | 800 | 0 | 0.5 | 5 |
| 4 | 1 | 0 | 0 | 0 | To failure |

**Table 2. Axial failure test results.**

| Osteotomy type | | Axial stiffness (N/mm) | | | Maximum strength (N) | | |
|---|---|---|---|---|---|---|---|
| | n | Mean | SD | p value | Mean | SD | p value |
| DCC | 3 | 1238.7 | 359.0 | 0.44 | 8175 | 516.9 | 0.36 |
| BP | 3 | 1277.3 | 144.4 | | 7832 | 1392.2 | |

## Results

The load-to-failure test results for axial and torsional loadings are shown in Table 1. There was no significant difference found between the two osteotomy techniques in terms of axial stiffness (1238.7 ± 359.0 vs. 1277.3 ± 144.4, p = 0.44) or maximum strength (8175 ± 516.9 vs. 7832 ± 1392.2, p = 0.36), with only a 4% difference in axial mechanical properties (Table 2).

In terms of torsional properties, the DCC configuration had a mean maximum torque 13% higher than the BP configuration. (63 ± 10.6 vs. 56 ± 12.1, p = 0.87). The DCC group also had significantly greater torsional stiffness compared to the BP group (27 ± 7.7 Nm/˚ vs. 4.5 ± 1.5 Nm/˚, p = 0.008) (Table 3).

### Failure modes in axial and rotational loading

No visible damage to the bone or implant was found at cyclic load stages prior to the failure test. After the final axial failure test, fractures were observed in both osteotomy configurations. In the BP configuration, the fracture occurred proximal to the hinge, specifically in the posterior part of the lateral femoral epicondyle. In the chevron configuration, the fracture occurred distally to the osteotomy site, propagating from the hinge to the condyle, which was embedded in polyurethane-based blocks (Fig 3A and 3B).

During the torsional failure tests, the BP configuration exhibited gross displacement at the osteotomy site due to the proximal femur rotating relative to the distal femur fragment. In the DCC configuration, no displacement at osteotomy site, while a spiral femur shaft bony fractures occurred proximal to the end of the locking plate, and only minimal displacement was observed at the osteotomy site (Fig 3C and 3D).

There were no instances of screw loosening, screw breakage, or bone plate damage observed during both axial and torsional loading modes.

### 3D model simulations for osteotomy contact surface

Our analysis of the contact surface area revealed that the single-plane osteotomy had a contact surface area of 1339 mm$^2$, while the BP osteotomy had a larger contact surface area of 2827 mm$^2$. In the BP model, the transverse cut had a contact surface area of 1129 mm$^2$, and the axial cut had a contact surface area of 1698 mm$^2$. The DCC osteotomy had a smaller contact surface area of 1749 mm$^2$, with two contact surfaces of 863 mm$^2$ and 886 mm$^2$, respectively (Fig 4).

**Table 3. Torsional failure test results.**

| Osteotomy type | | Torsional stiffness (Nm/˚) | | | Maximum torque (Nm) | | |
|---|---|---|---|---|---|---|---|
| | n | Mean | SD | p value | Mean | SD | p value |
| DCC | 3 | 27.0 | 7.7 | 0.008* | 63.0 | 10.6 | 0.87 |
| BP | 3 | 4.5 | 1.5 | | 56.0 | 12.1 | |

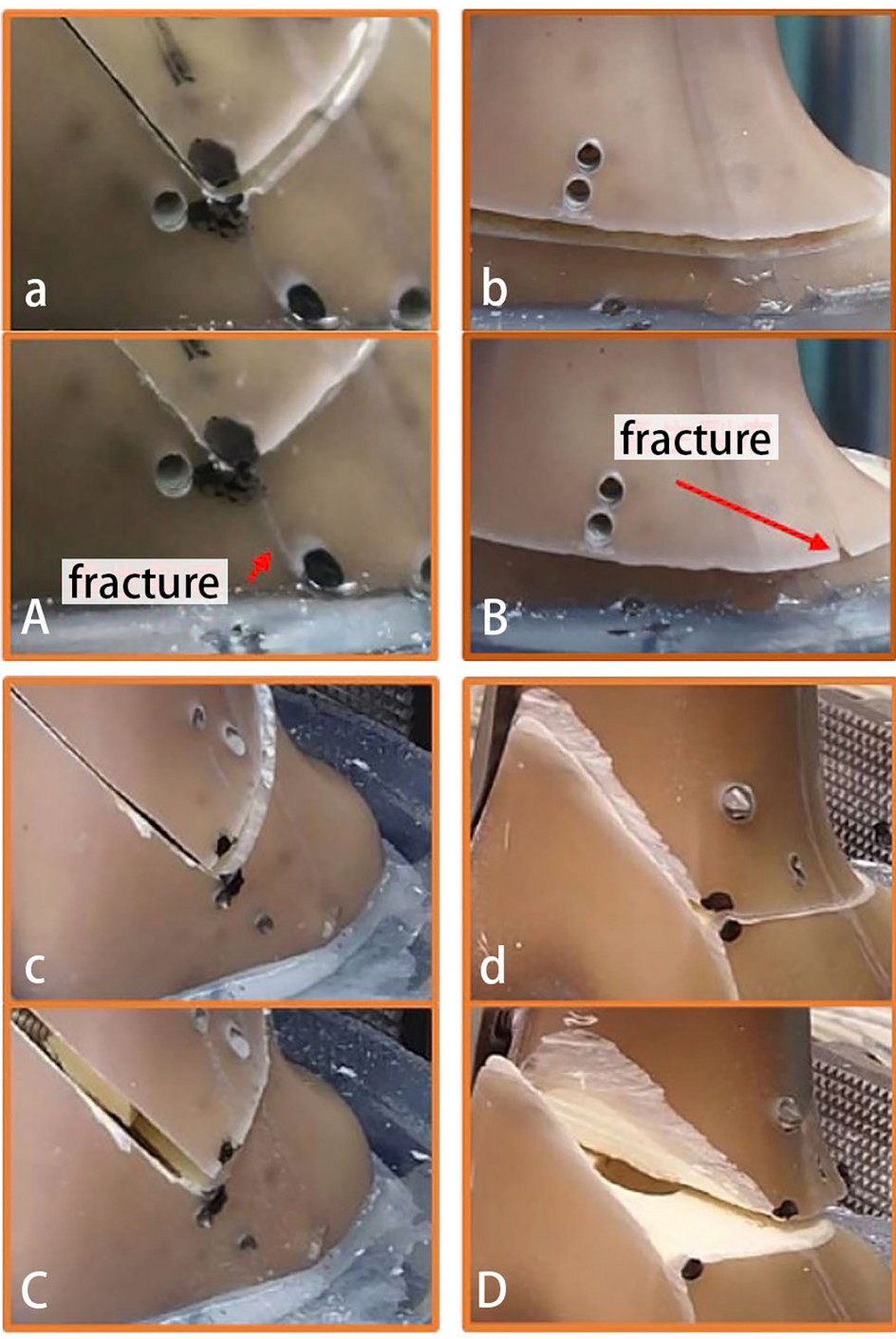

**Fig 3. Fracture patterns and displacement behaviors in BP and DCC configurations.** Axial Failure Test Results—DCC Configuration: Before the test, (a), and after the test, (A). Axial Failure Test Results—BP Configuration: Before the test, (b), and after the test, (B). Torsional Failure Test Results—DCC Configuration: Before the test, (c), and after the test, (C). No displacement at the osteotomy site was observed. Instead, a spiral femur shaft bony fracture occurred proximal to the end of the locking plate, with only minimal displacement observed at the osteotomy site. Torsional Failure Test Results—BP Configuration: Before the test, (d), and after the test, (D).

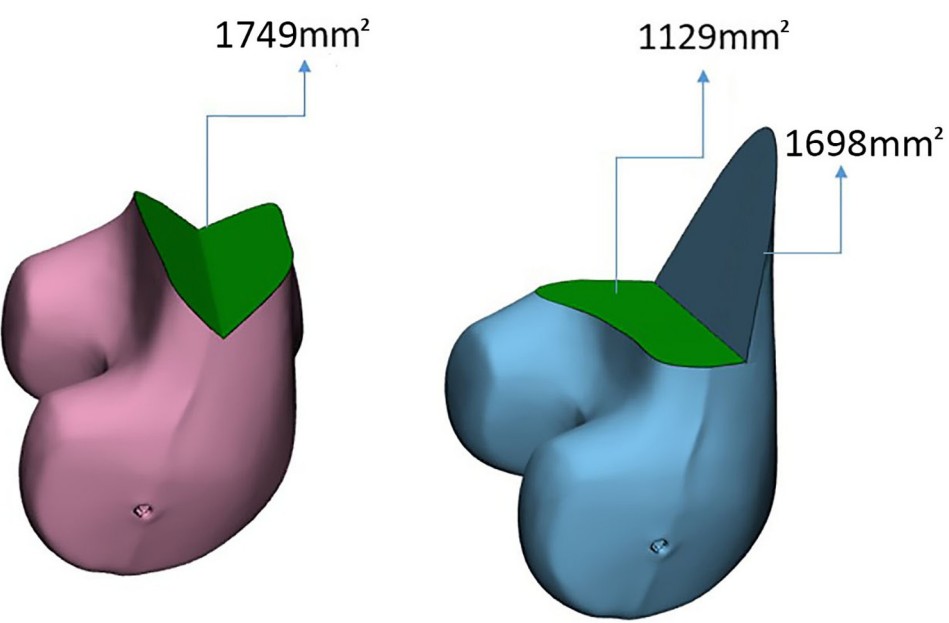

**Fig 4. 3D model analysis of contact surface area.** Left: DCC Model—The contact surface area measured 1749 mm$^2$. Right: BP Model—The transverse cut had a contact surface area of 1129 mm$^2$, and the axial cut had a contact surface area of 1698 mm$^2$.

## Discussion

Since DFO was first proposed in the 1980s, a variety of techniques, modifications, and implants have been suggested to improve the stability of construct, hinge protection, and accuracy of correction. The BP DFO technique introduced by Freiling et al. were among the most commonly utilized technique [4]. Additionally, the DCC technique presents itself as another appealing option [10].

Biomechanical studies have demonstrated the advantages of the BP technique, including smaller osteotomy wedge volume and larger contact surfaces compared to uniplanar DFO techniques, potentially leading to improved bone healing [16]. Furthermore, the Brinkman et al. study reported that the BP technique resulted in 2.7 times greater axial stability than the single-plane technique [5]. However, contrary to theoretical expectations, the same study found that torsional stability of BP DFO was 28% lower than that of the single-plane technique. This unexpected result may be attributed to the mechanically fragile anterior flange and the reduction in the hinge length on the lateral cortex [5,6]. In our study, we found that chevron construct was at least 600% stiffer than the BP construct. Rotational failure in BP osteotomy originated from the split at the hinge laterally, resulting in rotational displacement between proximal and distal end. In comparison, no displacement at the hinge was found in the DCC technique; the integrity of the osteotomy was maintained whereas failure occurred at midshaft facture.

The ultimate goal of a biomechanically improved DFO construct is to achieve immediate weight bearing [17]. Although immediate weight bearing had long been the postoperative care standard after high tibial osteotomy [18], most DFO series requiring 4–12 weeks of non- or partial weight-bearing postoperatively [19]. Van der Woude et al. compared the bone healing time in single-plane and BP techniques for distal femoral valgus osteotomy and found that the

mean bone healing time of BP osteotomies was 2 months shorter than that of uniplanar osteotomies (4 vs. 6 months) [20]. However, the time to full weight bearing was uniformly 6 weeks in both groups, with crutches required before 6 weeks. This suggests that BP osteotomy may improve bone healing clinically, but not necessarily mechanical stability. In contrast, in our previous clinical study of DCC DFO, patients were instructed to weight bear 'as tolerated' immediately postoperatively. This key aspect ensures the feasibility of immediate weight bearing in the DCC osteotomy procedure, as patients can gradually bear weight based on their comfort and pain tolerance. We found a mean time to full weight bearing of 3.7 weeks, with some cases even able to bear full weight as early as 2 weeks postoperatively. Union of the osteotomy was achieved in 11.3 weeks [10]. Additionally, it's worth noting that the accelerated timeline for weight bearing seen in our patients may be attributable to the superior mechanical stability offered by the DCC method.

As previous studies have suggested, increasing the cancellous bone surface area at the osteotomy site may facilitate accelerated contact healing and enhance the overall healing potential of osteotomies [16,21,22]. Van Heerwaarden's study showed that the contact surface of BP osteotomies, excluding the anterior flange, is slightly smaller than that of single-plane techniques. [16] Our 3D simulation showed similar results, demonstrating that the BP configuration has the largest contact surface area, 111% larger than that of the single-plane configuration. However, 60% of the contact surface area comes from the less reliable axial cut that creates mechanically fragile anterior flange. On the other hand, the DCC osteotomy offered a contact surface area 31% larger than that of the single-plane configuration, with both contact surfaces being weight-bearing. The surface area analysis corroborates our findings that chevron configuration exhibited greater stability than BP in our biomechanical analysis.

While the DCC configuration offers advantages in terms of rotational stiffness and early weight bearing, it also has significant limitations that need to be taken into consideration. One major limitation is the requirement for precise execution and the use of patient-specific instruments to ensure accuracy. The cut must be made at a specific location, specifically towards the upper border of the lateral femoral condyle, to create a perfect hinge point [23,24]. Our experience has shown that using the freehand technique for the DCC configuration can result in inadequate cuts that are unable to close the osteotomy, or overcuts that lead to hinge fracture.

On the other hand, the use of patient-specific simulations and personalized guides provides a reliable method to ensure unchanged rotation with DFO. In contrast, the orientation of bone cuts in BP DFO inevitably leads to unwanted rotational changes in the axial and sagittal planes [8,9]. This justifies the use of patient-specific instruments for the DCC configuration, as they help overcome the limitations associated with freehand techniques and ensure accurate and precise execution of the osteotomy [25–27].

Our study has several limitations that should be considered. Firstly, the use of synthetic bone models may not accurately reflect the properties of human bone, although the sawbone model has been made as mechanically close to human bone as possible. Secondly, the study only investigated two specific configurations of DFO and did not include the single plane osteotomy techniques. Thirdly, our study was conducted under controlled laboratory conditions, which may not reflect the complexity of in vivo conditions involving muscle forces, joint kinematics, and bone healing processes. Fourthly, our study employed the Instron E10000 system to record load, deflection, torque, and rotation, measuring motion across the specimen's mounting blocks. This method contrasts with Brinkman et al.'s direct measurement of osteotomy site motion using an ultrasound 3D motion-analysis system and femoral reference points. Finally, the sample size in our study was relatively small, with only six specimens tested for each osteotomy configuration. A larger sample size with more bone model varieties may have

provided more representative statistics and more precise estimates of the biomechanical properties of the osteotomy constructs.

## Conclusion

Our study found that DCC DFO configuration demonstrated significantly greater torsional stiffness compared to the BP configuration. 3D simulation analysis also demonstrated a 31% larger contact surface in comparison with single plane DFO. The results suggest that DCC configuration may offer better mechanical stability and have potential for earlier weight-bearing postoperatively.

## Supporting information

**S1 File.**
(PDF)

## Acknowledgments

The authors thank OFC Chang for providing language help and writing assistance to the manuscript.

## Author Contributions

**Conceptualization:** Ye Huang, Jesse Chieh-Szu Yang.

**Data curation:** Kuan-Jung Chen, Kuo-Kuang Huang, Jesse Chieh-Szu Yang.

**Formal analysis:** Kuan-Jung Chen.

**Investigation:** Kuan-Jung Chen, Chien-Yuan Wang.

**Methodology:** Chien-Yuan Wang, Jesse Chieh-Szu Yang.

**Project administration:** Chien-Yuan Wang, Oscar Kuang-Sheng Lee, Jesse Chieh-Szu Yang.

**Resources:** Wei Hsiung.

**Software:** Wei Hsiung, Kuo-Kuang Huang.

**Supervision:** Ye Huang, Jesse Chieh-Szu Yang.

**Visualization:** Oscar Kuang-Sheng Lee.

**Writing – original draft:** Kuan-Jung Chen, Wei Hsiung.

**Writing – review & editing:** Oscar Kuang-Sheng Lee, Kuo-Kuang Huang, Ye Huang, Jesse Chieh-Szu Yang.

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
