## [Decision Letter · Decision Letter 0]

4 Oct 2023

PONE-D-23-24697Comparison of Double Chevron-Cut and Biplanar Distal Femoral Osteotomy Techniques: A Biomechanical StudyPLOS ONE

Dear Dr. Yang,

Thank you for submitting your manuscript to PLOS ONE. After careful consideration, we feel that it has merit but does not fully meet PLOS ONE’s publication criteria as it currently stands. Therefore, we invite you to submit a revised version of the manuscript that addresses the points raised during the review process.

Please, address all the comments made by the reviewers. 

We look forward to receiving your revised manuscript.

Kind regards,

Antonio Riveiro Rodríguez, PhD

Academic Editor

PLOS ONE

Journal Requirements:

Reviewers' comments:

Reviewer's Responses to Questions

**Comments to the Author**

1. Is the manuscript technically sound, and do the data support the conclusions?

Reviewer #1: Yes

Reviewer #2: Partly

2. Has the statistical analysis been performed appropriately and rigorously? 

Reviewer #1: Yes

Reviewer #2: No

3. Have the authors made all data underlying the findings in their manuscript fully available?

Reviewer #1: Yes

Reviewer #2: Yes

4. Is the manuscript presented in an intelligible fashion and written in standard English?

Reviewer #1: Yes

Reviewer #2: Yes

5. Review Comments to the Author

Reviewer #1: Journal : PLOS ONE

Title : Comparison of Double Chevron-Cut and Biplanar Distal Femoral Osteotomy Techniques: A Biomechanical Study

ID : PONE-D-23-24697

Authors : Yang et al.

Manuscript Type : Research Original Paper

Date Reviewed : August 2023

Dear Editor,

I would like to express my gratitude for the opportunity to review the research paper under consideration for publication in PLOS ONE. The submitted paper has been meticulously examined and evaluated in terms of its scientific quality and language proficiency.

The paper presents a research study based on an experimental method, aiming to compare the stability and mechanical properties of the double chevron-cut (DCC) and biplanar (BP) distal femoral osteotomy (DFO) techniques, while also analysing their respective contact surface areas. The reviewer agrees that investigating these issues is crucial, and the topic aligns well with the interdisciplinary scope of the journal.

The abstract and main text effectively elucidate the research background. The originality, contribution to science, major objectives, methodology, discussion, and conclusion sections of the paper have been articulated. In particular, the research sequence is well-defined. However, it is worth noting that while the paper cites approximately 14 scientific references, it would greatly benefit from an expanded reference list to provide a more robust foundation for this study.

Based on the content of the research, I recommend the following revisions to enhance the paper's quality within a biomechanics perspective:

1. The paper should provide the dimensions of the model (Femur) used in the experimental test according to technical drawing standards.

2. Material properties necessary for engineering calculations, such as Young’s modulus, Poisson’s ratio, density, and yield or ultimate stress magnitudes, should be supplied for both the test material and the human bone structure. Furthermore, it should be compared to demonstrate how closely the tested model replicates real human bone properties. Additionally, an explanation of how the cortical and trabecular structures contribute to the mechanical aspects of this testing should be included.

3. While it is understood that surface area is a core parameter for force-based stress calculations to measure the failure state, the paper should elaborate on why the calculation of contact surface area was necessary for this study.

4. The capacity of the testing machine (Instron E10000 testing machine) should be clearly indicated.

5. The testing procedure section needs further clarification. It is suggested that schematic figures be added to illustrate each testing phase, such as axial and torsional loading in static or cyclic loading setups. It is understood that the experimental setup initially involves cyclic loading conditions and then determines the failure limits based on forces. However, the paper should provide a clearer explanation of why 100 cycles were used and how deformation at the end of the cyclic loading was assessed.

6. The rationale behind the choice of 100 cycles and loading frequencies of 0.25 and 0.5 Hz should be explained.

7. For Figure 3, the inclusion of a metric scale in the images would facilitate a visual comparison of failure and fracture magnitudes/masure.

8. Given that the paper cites approximately 14 scientific references, and while the core methodology is well-established, it is advisable to expand the reference list, ideally doubling it, to provide stronger support for the methodological background.

9. As a reminder, the reviewer suggests that the paper could benefit from more professional graphic design approaches to enhance the quality of graphs, schematics, and tables. Ensure that letters, characters, and units in the figures and tables are consistent with those used in the main text and are easily readable. Additionally, test printouts should be presented with their test order, including clear numerical values and units, along with legible legends.

In conclusion, AFTER the suggested MAJOR REVISIONS ARE MADE, I RECOMMEND this paper for publication in PLOS ONE.

Yours sincerely,

Reviewer #2: (General comments)

In this paper, the authors describe the results of a biomechanical study using synthetic bone models and 3D image analysis to compare the properties of double Chevron-cut (DCC) and biplanar (BP) distal femoral osteotomy techniques.

Overall, the thread of the description is clearly organized, and the presented results showing mechanical advantage of DCC technique over BP technique provide clinically significant information adding to our current knowledge.

Reading through the context, however, there are some areas requiring further clarification as stated below.

I would suggest the authors to revise the contents of the manuscript by responding to the comments given.

(Specific comments)

1. Abstract

Line 30-31: The present study results may not be robust enough to draw these conclusions regarding advantages of the DCC technique for weight-bearing and bone healing over the BP technique.

2. Introduction

In general, the rationale and the purpose of this study are clearly stated in this section.

Please state the study hypothesis at the end of this section.

3. Materials and methods

Lines 100-103: Please describe how the accurate and reproducible alignment of the specimen was controlled and attained in the testing.

Lines 123-125: It is not clear how the 0-, 150-, and 800-N axial preloads were increasingly applied in the cyclical torsional loading test. Please present a table showing the test protocol (sequence).

Lines 129-130: In the previous relevant studies (References # 4 and 9), the mechanical behavior of the bone-implant construct during the torsional testing was assessed by the direct measurement of the motion at the osteotomy site using a 3-D motion analysis system, while the motion was measured by the material testing system (based on the motion between the mounting blocks at the end of the specimen) in this study.

The authors might like to refer to this difference in the assessment method of motion between the Brinkman’s study and the present study, and mention this issue in the limitations paragraph.

Line 134: Does the term “replicate femur” indicate the Sawbone model?

Statistical analysis: A parametric test (t-test) was used in the analysis. Did the authors confirm the feasibility (validity) to use this test in terms of the sample size and the data distribution?

4. Results

The obtained results are clearly presented in this section.

5. Discussion

Lines 214-227: The issue of early (immediate) weight bearing after surgery is discussed in this paragraph; however, the present study may not provide enough data to confirm the advantage of the DCC technique in timing of weight bearing as compared to the BP technique.

Please limit the statements to those supported by the concrete study results.

Lines 252-262 (Study limitations): Please consider referring to the method of measuring motion in the mechanical testing as commented above.

Line 260: The number of the specimen for each osteotomy technique tested for each loading condition was 3, and as stated here, this issue represents a study weakness.

6. Figures

Figures are clearly prepared to help the readers’ understanding.

7. Tables

As commented above, preparation of a table showing the test protocol (sequence) is suggested.

6. PLOS authors have the option to publish the peer review history of their article (what does this mean?). If published, this will include your full peer review and any attached files.

Reviewer #1: No

Reviewer #2: No

---

## [Author Response · Author response to Decision Letter 0]

14 Nov 2023

Dear Reviewers,

Thank you for your detailed feedback on our manuscript. We have revised the manuscript and tables according to your suggestions. Please find our responses to your comments below:

Thank you for your valuable input, which has greatly improved our manuscript. We hope that the revised version meets the standards required for publication in "PLOS ONE."

Reviewer #1:

Q1 & 2: 

The paper should provide the dimensions of the model (Femur) used in the experimental test according to technical drawing standards. Material properties necessary for engineering calculations, such as Young’s modulus, Poisson’s ratio, density, and yield or ultimate stress magnitudes, should be supplied for both the test material and the human bone structure. Furthermore, it should be compared to demonstrate how closely the tested model replicates real human bone properties. Additionally, an explanation of how the cortical and trabecular structures contribute to the mechanical aspects of this testing should be included.

Author’s Response: 

Thank you for your suggestions. For our experimental tests, we utilized a medium-sized femur sawbone (model #3403) with the following dimensions according to technical drawing standards: a total length of 455 mm, a neck-shaft angle of 135°, a shaft diameter of 27 mm, and a 17 PCF solid foam cancellous core with a 13 mm canal diameter 1. We have included the description in the manuscript. (Line 80-84)

Regarding the material properties of this model and its comparison to human bone structure, we acknowledge that these aspects are crucial for engineering calculations and replicating real human bone properties. However, we would like to highlight that the comparisons fall outside the scope of our current study. These properties have been previously detailed in well-established studies by Heiner AD 2,3. These studies extensively evaluated the stiffness of natural bone and sawbone under various loading conditions. We had included these references and revised the manuscript. (Line 83-84)

References:

1. Sawbone Biomechanical Products Catalog, Test Materials and Composite Bones. Sawbone. 2020 [Cited 2023 November 6]. Available from: https://www.sawbones.com/media/assets/product/documents/biomechanical_catalog2020.pdf

2. Heiner AD. Structural properties of fourth-generation composite femurs and tibias. J Biomech. 2008 Nov 14;41(15):3282-4. doi: 10.1016/j.jbiomech.2008.08.013. Epub 2008 Oct 1. PMID: 18829031; 

3. Heiner AD, Brown TD. Structural properties of a new design of composite replicate femurs and tibias. J Biomech. 2001 Jun;34(6):773-81. doi: 10.1016/s0021-9290(01)00015-x. PMID: 11470115).

Q3: 

While it is understood that surface area is a core parameter for force-based stress calculations to measure the failure state, the paper should elaborate on why the calculation of contact surface area was necessary for this study.

Author’s Response: 

While the impact of contact surface area on force distribution and stress within bone models is indeed an interesting topic for investigation, our primary objective in calculating the contact surface area was to assess its potential clinical influence on the healing process of osteotomies. As studies have suggested, increasing the cancellous bone surface area at the osteotomy site may facilitate accelerated contact healing and enhance the overall healing potential of osteotomies [1,2,3]

We have included this clarification and the relevant references in the revised manuscript. (Line 52, 251-253)

References:

1. van Heerwaarden R, Najfeld M, Brinkman M, Seil R, Madry H, Pape D. Wedge volume and osteotomy surface depend on surgical technique for distal femoral osteotomy. Knee Surg Sports Traumatol Arthrosc. 2013 Jan;21(1):206-12. doi: 10.1007/s00167-012-2127-y. Epub 2012 Jul 6. PMID: 22766687.

2. Pape D, Dueck K, Haag M, Lorbach O, Seil R, Madry H. Wedge volume and osteotomy surface depend on surgical technique for high tibial osteotomy. Knee Surg Sports Traumatol Arthrosc. 2013 Jan;21(1):127-33. doi: 10.1007/s00167-012-1913-x. PMID: 22293899.

3. Nejima S, Kumagai K, Fujimaki H, Yamada S, Sotozawa M, Matsubara J, Inaba Y. Increased contact area of flange and decreased wedge volume of osteotomy site by open wedge distal tibial tuberosity arc osteotomy compared to the conventional technique. Knee Surg Sports Traumatol Arthrosc. 2021 Oct;29(10):3450-3457. doi: 10.1007/s00167-020-06296-8. Epub 2020 Sep 28. PMID: 32986149.

Q4:

The capacity of the testing machine (Instron E10000 testing machine) should be clearly indicated.

Author’s Response: 

The Instron E10000 system boasts a linear dynamic capacity of 10 kN, torsional dynamic capacity of 100 Nm, and offers a weighing accuracy of ±0.5% for both load and torque measurements. We collected load and deflection data as well as torque and rotation angle data using the Instron E10000 system. We have included the description in the manuscript. (Line 117-121)

Q5 & 6: 

The testing procedure section needs further clarification. It is suggested that schematic figures be added to illustrate each testing phase, such as axial and torsional loading in static or cyclic loading setups. It is understood that the experimental setup initially involves cyclic loading conditions and then determines the failure limits based on forces. However, the paper should provide a clearer explanation of why 100 cycles were used and how deformation at the end of the cyclic loading was assessed.

The rationale behind the choice of 100 cycles and loading frequencies of 0.25 and 0.5 Hz should be explained.

Author’s Response: 

Thank you for your suggestions. We have included a Table Overview of the axial and torsional test protocols to clarify our loading test setup and protocols.

The choice of 100 cycles and loading frequencies of 0.25 and 0.5 Hz was influenced by previous studies with similar experimental setups, such as the works by Brinkman et al. [1,2] They decided that test runs had to be limited to 100 cycles for practical reasons, as data storage requirements would otherwise be too high. Our choice was made to ensure the generation of comparable data between our study and these previous works.

In our testing procedure, we collected load and deflection data, as well as torque and rotation angle data, using the Instron E10000 system. The details of these procedures have been included in the manuscript for clarity. (Table 1, Line 120-121)

References:

1. Brinkman JM, Hurschler C, Staubli AE, van Heerwaarden RJ. Axial and torsional stability of an improved single-plane and a new bi-plane osteotomy technique for supracondylar femur osteotomies. Knee Surg Sports Traumatol Arthrosc. 2011;19(7):1090-8.

2. Brinkman JM, Hurschler C, Agneskirchner JD, Freiling D, van Heerwaarden RJ. Axial and torsional stability of supracondylar femur osteotomies: biomechanical comparison of the stability of five different plate and osteotomy configurations. Knee Surg Sports Traumatol Arthrosc. 2011;19(4):579-87.

Q7: For Figure 3, the inclusion of a metric scale in the images would facilitate a visual comparison of failure and fracture magnitudes/measure.

Author’s Response: 

We appreciate the suggestion to include metric scales in Figure 3 to facilitate visual comparisons. Unfortunately, the photographs have already been taken, and it is not possible to add metric scales to the figure at this stage of the manuscript preparation. However, we will ensure that future figures in our research work include metric scales for improved clarity and accuracy. Thank you for the valuable feedback. 

Q8: Given that the paper cites approximately 14 scientific references, and while the core methodology is well-established, it is advisable to expand the reference list, ideally doubling it, to provide stronger support for the methodological background.

Author’s Response: 

Thank you for your suggestions. We have expanded the reference list to include more relevant studies, resulting in a total of 27 references. We believe this will provide stronger support for the methodological background.

Q9: As a reminder, the reviewer suggests that the paper could benefit from more professional graphic design approaches to enhance the quality of graphs, schematics, and tables. Ensure that letters, characters, and units in the figures and tables are consistent with those used in the main text and are easily readable. Additionally, test printouts should be presented with their test order, including clear numerical values and units, along with legible legends.

Author’s Response: 

We appreciate the reviewer's feedback regarding the visual elements of our paper. We acknowledge the importance of consistency, readability, and clarity in our figures, schematics, and tables. We will ensure that all elements are presented uniformly and that numerical values and units are clearly displayed. Additionally, we will work on providing informative and legible legends for our figures and tables to enhance the overall quality of our visuals. Thank you for these valuable suggestions, and we will make the necessary improvements to meet these criteria.

 

Reviewer #2:

Q1: Abstract Line 30-31: 

The present study results may not be robust enough to draw these conclusions regarding advantages of the DCC technique for weight-bearing and bone healing over the BP technique.

Author’s Response: 

We have revised the conclusions to present the pure factual findings from our study results, avoiding any speculative statements. The revised paragraph is as follows:

“The DCC osteotomy exhibited superior mechanical stability, showing improved rotational stiffness and maximum torque when compared to the BP osteotomy. Although the BP osteotomy resulted in a larger contact surface area than the DCC osteotomy, both were larger than the conventional single-plane configuration. In clinical practice, both the DCC and BP techniques should be evaluated based on patient-specific characteristics and surgical goals.” (Line 29-34)

Q2: Introduction

In general, the rationale and the purpose of this study are clearly stated in this section.

Please state the study hypothesis at the end of this section.

Author’s Response: 

Thank you for the comment. the study hypothesis had been added to the section. with its contents as follows:

"We hypothesize that the DCC technique will demonstrate superior mechanical stability and stiffness compared to the BP DFO technique under both axial and rotational loads. Additionally, we expect to observe differences in the cutting site and contact surface area among the three osteotomy techniques (single plane, DCC, and BP)." (Line 72-76)

Q3: Materials and methods

Lines 100-103: 

Please describe how the accurate and reproducible alignment of the specimen was controlled and attained in the testing.

Author’s Response: 

We appreciate the reviewer's question and would like to clarify that the accurate and reproducible alignment of the specimens in our study was achieved during the initial setup of the experiments. We ensured that the femur specimens were securely mounted on the testing machine (Instron E10000) in a valgus position to mimic normal weight-bearing conditions of the lower limb. The line of action for axial compression load was established to pass through the center of the femoral head and the intercondylar notch, simulating the mechanical axis of the femur. Importantly, once the initial setup was completed, no specialized tools or further adjustments were used during the testing process. This approach was chosen to maintain consistency and avoid introducing variability in the alignment of the specimens during testing. We hope this clarification addresses the reviewer's query regarding specimen alignment.

Lines 123-125: 

It is not clear how the 0-, 150-, and 800-N axial preloads were increasingly applied in the cyclical torsional loading test. Please present a table showing the test protocol (sequence).

Author’s Response: 

Thank you for your suggestions. We have included a Table Overview of the axial and torsional test protocols to clarify our loading test setup and protocols. (Table 1 Overview of the test protocols)

Lines 129-130: 

In the previous relevant studies (References # 4 and 9), the mechanical behavior of the bone-implant construct during the torsional testing was assessed by the direct measurement of the motion at the osteotomy site using a 3-D motion analysis system, while the motion was measured by the material testing system (based on the motion between the mounting blocks at the end of the specimen) in this study.

The authors might like to refer to this difference in the assessment method of motion between the Brinkman’s study and the present study, and mention this issue in the limitations paragraph.

Author’s Response: 

We appreciate your observation. Indeed, this distinction in measurement methodology is a significant one. In our current study, we have used the Instron E10000 system to acquire and record data on load, deflection, as well as torque and rotation angle, considering the motion between the mounting blocks at the specimen's ends. This differs from the approach taken by Brinkman et al., where the motion at the osteotomy site was directly measured relative to specific reference points on the femur, using an ultrasound 3D motion-analysis system. We have acknowledged this difference in our discussion and limitations section, reflecting on how it may influence the comparison of results across studies. Your comment underscores the importance of this acknowledgment, and we will ensure it is clearly articulated. (Line 121-122, 286-290)

Line 134: 

Does the term “replicate femur” indicate the Sawbone model?

Author’s Response: 

Yes, the sentence was referring to the sawbone femur model. We corrected the term “replicate femur” to "sawbone femur" in order to clarify the expression. Thank you for pointing out. (Line 9 and 125)

Statistical analysis: 

A parametric test (t-test) was used in the analysis. Did the authors confirm the feasibility (validity) to use this test in terms of the sample size and the data distribution?

Author’s Response: 

Thank you for your inquiry. Prior to utilizing the parametric t-test in our analysis, we confirmed data normality with the Shapiro-Wilk test. This validation step ensures the appropriateness of the t-test for our dataset. We have included this procedural detail in the revised manuscript to clarify our analytical approach. (Line 164-165)

Q4: Results

The obtained results are clearly presented in this section.

Author’s Response: 

We appreciate your feedback on the results presentation. Your positive assessment encourages us to maintain clarity in our manuscript.

Q5: Discussion

Lines 214-227: 

The issue of early (immediate) weight bearing after surgery is discussed in this paragraph; however, the present study may not provide enough data to confirm the advantage of the DCC technique in timing of weight bearing as compared to the BP technique. Please limit the statements to those supported by the concrete study results.

Author’s Response: 

Thank you for your comment. Our current biomechanical study aims to provide a rationale for the early full weight bearing observed with the DCC DFO technique, as reported in our prior clinical research. The accelerated timeline for weight bearing seen in our patients, averaging 3.7 weeks compared to the 4-12 weeks noted in the existing literature, may be attributable to the superior mechanical stability offered by the DCC method. We believe our discussion appropriately contextualizes our biomechanical data within the broader scope of clinical application and will ensure this connection is made explicit to support our conclusions.

Lines 252-262 (Study limitations): 

Please consider referring to the method of measuring motion in the mechanical testing as commented above.

Author’s Response: 

Thank you for pointing out. We had amended our manuscript to detail our motion measurement method and contrast it with previous approaches, as suggested. Thank you for the guidance. (Line 286-290)

Line 260: 

The number of the specimen for each osteotomy technique tested for each loading condition was 3, and as stated here, this issue represents a study weakness.

Author’s Response: 

The sample size in our study was indeed relatively small, with only six specimens tested for each osteotomy configuration. we acknowledged this as a study weakness 

Q6: Figures

Figures are clearly prepared to help the readers’ understanding.

Author’s Response: 

Thank you for acknowledging the clarity of our figures. We are pleased that our visual aids are effectively assisting readers' comprehension.

Q7: Tables

As commented above, preparation of a table showing the test protocol (sequence) is suggested.

Author’s Response:

We appreciate your recommendations. In response, we have incorporated a table (Table 1: Overview of Test Protocols) to provide a clearer presentation of our loading test setup and protocols.

---

## [Decision Letter · Decision Letter 1]

24 Nov 2023

PONE-D-23-24697R1Comparison of Double Chevron-Cut and Biplanar Distal Femoral Osteotomy Techniques: A Biomechanical StudyPLOS ONE

Dear Dr. Yang,

Thank you for submitting your manuscript to PLOS ONE. After careful consideration, we feel that it has merit but does not fully meet PLOS ONE’s publication criteria as it currently stands. Therefore, we invite you to submit a revised version of the manuscript that addresses the points raised during the review process. Please, address all the comments made by the reviewers, and highlight all the changes in the revised manuscript.  Please submit your revised manuscript by Jan 08 2024 11:59PM. If you will need more time than this to complete your revisions, please reply to this message or contact the journal office at plosone@plos.org. Please include the following items when submitting your revised manuscript:A rebuttal letter that responds to each point raised by the academic editor and reviewer(s). You should upload this letter as a separate file labeled 'Response to Reviewers'.A marked-up copy of your manuscript that highlights changes made to the original version. You should upload this as a separate file labeled 'Revised Manuscript with Track Changes'.An unmarked version of your revised paper without tracked changes. You should upload this as a separate file labeled 'Manuscript'.If applicable, we recommend that you deposit your laboratory protocols in protocols.io to enhance the reproducibility of your results. Protocols.io assigns your protocol its own identifier (DOI) so that it can be cited independently in the future. For instructions see: https://journals.plos.org/plosone/s/submission-guidelines#loc-laboratory-protocols. Additionally, PLOS ONE offers an option for publishing peer-reviewed Lab Protocol articles, which describe protocols hosted on protocols.io. Read more information on sharing protocols at https://plos.org/protocols?utm_medium=editorial-email&utm_source=authorletters&utm_campaign=protocols.

We look forward to receiving your revised manuscript.

Kind regards,

Antonio Riveiro Rodríguez, PhD

Academic Editor

PLOS ONE

Journal Requirements:

Reviewers' comments:

Reviewer's Responses to Questions

**Comments to the Author**

1. If the authors have adequately addressed your comments raised in a previous round of review and you feel that this manuscript is now acceptable for publication, you may indicate that here to bypass the “Comments to the Author” section, enter your conflict of interest statement in the “Confidential to Editor” section, and submit your "Accept" recommendation.

Reviewer #1: All comments have been addressed

Reviewer #2: (No Response)

2. Is the manuscript technically sound, and do the data support the conclusions?

Reviewer #1: (No Response)

Reviewer #2: Partly

3. Has the statistical analysis been performed appropriately and rigorously? 

Reviewer #1: N/A

Reviewer #2: Yes

4. Have the authors made all data underlying the findings in their manuscript fully available?

Reviewer #1: Yes

Reviewer #2: Yes

5. Is the manuscript presented in an intelligible fashion and written in standard English?

Reviewer #1: Yes

Reviewer #2: Yes

6. Review Comments to the Author

Reviewer #1: Dear Editor,

I am of the opinion that the revised version of this paper meets the standards for publication. I would like to propose its consideration for publication in its current revised format.

Regards,

Reviewer #2: In this revised version of the manuscript and the document describing the authors’ response to the reviewer, the authors have appropriately responded to each of the comments given. Consequently, the clarity of the description has been improved; however, there are a few areas requiring further clarification as described below.

Lines 112-118: Please describe in more detail regarding how the specimen was positioned and aligned to mimic the normal weight bearing.

Lines 238-251: It is still not clear enough which aspect of the study results can ensure the feasibility of immediate weight bearing in DCC osteotomy procedure.

7. PLOS authors have the option to publish the peer review history of their article (what does this mean?). If published, this will include your full peer review and any attached files.

Reviewer #1: No

Reviewer #2: No

---

## [Author Response · Author response to Decision Letter 1]

29 Nov 2023

Dear Reviewers,

Thank you for your detailed feedback on our manuscript, which has greatly improved our manuscript. We have revised the manuscript and tables according to your suggestions. We hope that the revised version meets the standards required for publication in "PLOS ONE." Please find our responses to your comments below:

Reviewer #1:

I am of the opinion that the revised version of this paper meets the standards for publication. I would like to propose its consideration for publication in its current revised format.

Author’s Response: 

Thank you for your positive feedback on our revised manuscript. We appreciate your support for its consideration for publication in its current form. Your insights have been invaluable in improving our work.

We look forward to the possibility of our paper being published in PLOS ONE.

Reviewer #2:

In this revised version of the manuscript and the document describing the authors’ response to the reviewer, the authors have appropriately responded to each of the comments given. Consequently, the clarity of the description has been improved; however, there are a few areas requiring further clarification as described below.

Q1: Lines 112-118

Please describe in more detail regarding how the specimen was positioned and aligned to mimic the normal weight bearing.

Author’s Response: 

We appreciate the reviewer's feedback and their request for more detailed information on how the specimen was positioned and aligned to mimic normal weight bearing. In response to this valuable suggestion, we have revised the manuscript to provide a more comprehensive description of the alignment procedure, which we believe will enhance the clarity and understanding of our experimental setup. We hope that these additions address the reviewer's concern adequately. Thank you for your constructive comments. (Line 110-115)

Q2: Lines 238-251

It is still not clear enough which aspect of the study results can ensure the feasibility of immediate weight bearing in DCC osteotomy procedure.

Author’s Response: 

We appreciate the reviewer's feedback. The biomechanical study focused on assessing mechanical stability and stiffness. The connection between the study results and immediate weight bearing in the DCC osteotomy procedure lies in the hypothesis that greater mechanical stability, particularly superior rotational stiffness observed in the DCC group, may enable patients to initiate early weight-bearing based on comfort and pain tolerance. This aspect contributes to the clinical outcome observed in our previous study with a mean time to full weight bearing of 3.7 weeks. We have clarified this connection in the revised manuscript. (Line 244-253)

---

## [Decision Letter · Decision Letter 2]

10 Dec 2023

Comparison of Double Chevron-Cut and Biplanar Distal Femoral Osteotomy Techniques: A Biomechanical Study

PONE-D-23-24697R2

Dear Dr. Yang,

We’re pleased to inform you that your manuscript has been judged scientifically suitable for publication and will be formally accepted for publication once it meets all outstanding technical requirements.

Kind regards,

Antonio Riveiro Rodríguez, PhD

Academic Editor

PLOS ONE

Reviewers' comments:

Reviewer's Responses to Questions

**Comments to the Author**

1. If the authors have adequately addressed your comments raised in a previous round of review and you feel that this manuscript is now acceptable for publication, you may indicate that here to bypass the “Comments to the Author” section, enter your conflict of interest statement in the “Confidential to Editor” section, and submit your "Accept" recommendation.

Reviewer #2: All comments have been addressed

2. Is the manuscript technically sound, and do the data support the conclusions?

Reviewer #2: Yes

3. Has the statistical analysis been performed appropriately and rigorously? 

Reviewer #2: Yes

4. Have the authors made all data underlying the findings in their manuscript fully available?

Reviewer #2: Yes

5. Is the manuscript presented in an intelligible fashion and written in standard English?

Reviewer #2: Yes

6. Review Comments to the Author

Reviewer #2: (No Response)

7. PLOS authors have the option to publish the peer review history of their article (what does this mean?). If published, this will include your full peer review and any attached files.

Reviewer #2: No

---

## [Editor Report · Acceptance letter]

3 Apr 2024

PONE-D-23-24697R2 

PLOS ONE

Dear Dr. Yang, 

I'm pleased to inform you that your manuscript has been deemed suitable for publication in PLOS ONE. Congratulations! Your manuscript is now being handed over to our production team.

Kind regards, 

on behalf of

Dr. Antonio Riveiro Rodríguez 

Academic Editor

PLOS ONE